# Early Outpatient Treatment of COVID-19: A Retrospective Analysis of 392 Cases in Italy

**DOI:** 10.3390/jcm11206138

**Published:** 2022-10-18

**Authors:** Marco Cosentino, Veronica Vernocchi, Stefano Martini, Franca Marino, Barbara Allasino, Maria Antonietta Bàlzola, Fabio Burigana, Alberto Dallari, Carlo Servo Florio Pagano, Antonio Palma, Mauro Rango

**Affiliations:** 1Center for Research in Medical Pharmacology, University of Insubria, 21100 Varese, Italy; 2IppocrateOrg Association, Via Canova 15, 6900 Lugano, Switzerland

**Keywords:** SARS-CoV-2, COVID-19, primary care, family medicine, early outpatient treatment

## Abstract

COVID-19 was declared a pandemic in March 2020. The knowledge of COVID-19 pathophysiology soon provided a strong rationale for the early use of both anti-inflammatory and antithrombotic drugs; however, its evidence was slowly and partially incorporated into institutional guidelines. The unmet needs of COVID-19 outpatients were taken care of by networks of physicians and researchers. We analyse the characteristics, management and outcomes in COVID-19 outpatients who were taken care of by physicians within the IppocrateOrg Association. In this observational retrospective study, volunteering doctors provided data on 392 COVID-19 patients. The mean age of patients was 48.5 years (range: 0.5–97), and patients were taken care of in COVID-19 stage 0 (15.6%), stage 1 (50.0%), stage 2a (28.8%) and stage 2b (5.6%). Many patients were overweight (26%) or obese (11.5%), with chronic comorbidities (34.9%), mainly cardiovascular (23%) and metabolic (13.3%). The most frequently prescribed drugs included: vitamins and supplements (98.7%), aspirin (66.1%), antibiotics (62%), glucocorticoids (41.8%), hydroxychloroquine (29.6%), enoxaparin (28.6%), colchicine (8.9%), oxygen therapy (6.9%), and ivermectin (2.8%). Hospitalization occurred in 5.8% of cases, mainly in stage 2b (27.3%). A total of 390 patients (99.6%) recovered; one patient was lost at follow up, and one patient died after hospitalization. This is the first real-world study describing the behaviours of physicians caring for COVID-19 outpatients, and the outcomes of COVID-19 early treatment. The lethality in this cohort was 0.2%, while overall, and over the same period, the COVID-19 lethality in Italy was over 3%. The drug use described in this study appears effective and safe. The present evidence should be carefully considered by physicians and political decision makers.

## 1. Introduction

An outbreak of pneumonia due to the novel severe acute respiratory syndrome (SARS)-coronavirus-2 (CoV-2) emerged in December 2019 in the central Chinese province of Hubei [1]. Quickly spreading around the world with apparently high contagiousness and lethality, the coronavirus disease 2019 (COVID-19) was eventually declared a pandemic by the World Health Organization in March 2020 [2]. By that time, according to official data, Italy had 12,462 confirmed cases and 827 deaths, with a case fatality rate of 6.64%. China was the only country with a higher number of recorded deaths due to this COVID-19 outbreak [3].

The initial approach by Italian Health Authorities was to contain the spread of SARS-CoV-2; the COVID-19 emergency response was based essentially on social distancing, isolation and the virological testing of patients, with the tracing and quarantine of asymptomatic contacts. Even autopsies on subjects whose deaths were caused by COVID-19 were discouraged, due to the fear of spreading the virus [4]. Fortunately, despite such recommendations, seminal studies were performed, which led to the detailed description of the predominant pattern of inflammatory lung lesions in patients with COVID-19, including diffuse alveolar damage, hyaline membrane formation, atypical pneumocyte hyperplasia and frequent extensive coagulopathy [5], thus providing a strong rationale for the early use of anti-inflammatory, antiplatelet and anticoagulant drugs in COVID-19.

Unfortunately, the evidence of these studies was only partially incorporated into various versions of the Italian Health Authorities’ guidelines for the early treatment of patients with the SARS-CoV-2 infection, until even the latest version [6], which recommends only the symptomatic treatment of high body temperature with paracetamol and possibly non-steroidal anti-inflammatory drugs (NSAID), while warning against extensive use of glucocorticoids and low molecular weight heparins, and advising against antibiotics, hydroxychloroquine and other antivirals. At a certain point, the Italian drugs agency, AIFA, also denied the authorization of the off-label use of hydroxychloroquine in COVID-19 [7], a decision which conflicted with the right of doctors to prescribe therapeutics in the best interest of their individual patients, which is a doctor’s exclusive prerogative, and guaranteed by law.

Against this context, the IppocrateOrg Association, an international network of physicians, researchers, health and social workers developed evidence-based guidelines for the early outpatient treatment of COVID-19. These guidelines were based on: the international experience gained by doctors caring for COVID-19 patients from the very beginning of the pandemic in January 2020; the valuable experiences of clinicians, pneumologists and infectious disease specialists who, for decades, took care of patients with similar infectious diseases leading to interstitial lung disease; and doctors’ empirical experience of facing the COVID-19 epidemic in Italy. The resulting guidelines are regularly updated by doctors volunteering within the assistance network, who share and discuss their experiences with colleagues and leading experts, and who consider novel and emerging medical and scientific evidence. Such guidelines represent a non-binding reference for volunteering doctors. The suggested prescriptions are listed with suggested dosages, warnings and contraindications, according to the main COVID-19 stages [8,9,10]: stage 0 (asymptomatic with positive swab), stage 1 (symptomatic without signs of lung disease), and stages 2a and 2b (symptomatic with lung disease without and with desaturation). Doctors ultimately choose prescriptions that are in the best interest of patients, with appropriate doses and therapeutic regimens, considering the patient’s age, the COVID-19 stage and severity, the presence of relevant comorbidities and/or other risk factors (Appendix A).

The present study represents the first attempt to provide a comprehensive description and analysis of COVID-19 patients who were cared for by physicians volunteering within the IppocrateOrg Association assistance network. Section 2 provides a description of applied assistance procedures. The reported results describe the care and outcomes of nearly 400 COVID-19 patients who were assisted in Italy from 1 November 2020 to 31 March 2021, during the “second wave” of COVID-19. The outcomes are assessed in terms of recovery, without or with sequelae, hospitalization and death. 

## 2. Materials and Methods

### 2.1. Setting

In October 2020, IppocrateOrg instituted a service of medical teleassistance to better manage the “second wave”. IppocrateOrg doctors developed specific approaches for the prevention and early treatment of COVID-19. The extreme importance of timely treatment, received within the first few days from the emergence of initial symptoms, soon became evident. Throughout the course of the “second wave”, thousands of people were assisted and treated by doctors through teleassistance. In March 2021, the book *Guarire il COVID a Casa*, written by Mauro Rango and the group of volunteer doctors [11], was published by IppocrateOrg, becoming the first manual for specific home therapies.

IppocrateOrg’s assistance procedures are available on the web (https://ippocrateorg.org/assistenza-999/, accessed on 3 October 2022) and may be accessed by anyone who is seeking help. The service is completely free to its users [11].

### 2.2. Study Procedures

We performed an observational retrospective study to analyse the characteristics, management and outcomes of a sample of COVID-19 patients taken care of by physicians volunteering within the assistance network that is promoted and supported by the IppocrateOrg Association. The observation period was chosen from 1 November 2020, thus one month after the institution of the IppocrateOrg Association service of medical teleassistance, to allow for an initial run-in period, to 31 March 2021, to include the bulk of the second wave of COVID-19.

All doctors volunteering during the selected period were invited to provide the following information regarding the COVID-19 patients they consecutively cared for: the disease stage when patient care commenced (according to the modified staging system based on Siddiqi and Mehra (2020) and Turk et al. (2020) [8,9]); the demographic and anthropometric data; chronic comorbidities; the outcome of nasopharyngeal swabs; and prescribed drugs. All information was previously collected by physicians as a routinary part of their professional activity and provided in anonymized form. The outcomes of the study were hospitalization, recovery, without or with sequelae, and death.

### 2.3. Questionnaire for Participating Doctors

To gain further insight into the attitudes of individual doctors concerning the choice, use and the possible adverse drug reactions (ADR) of drugs included in the IppocrateOrg Association guidelines, we asked participating doctors to complete a questionnaire investigating their propensity to prescribe each drug in the various stages of COVID-19, and the relevant factors which eventually affected their decision (see Appendix B).

### 2.4. Statistical Analysis

Data was collected by means of an electronic datasheet (Microsoft Excel Version 2209, Microsoft Corporation, Redmond, WA, USA, https://office.microsoft.com/excel) and analysed by descriptive statistics. The differences between continuous variables were assessed via one-way ANOVA for independent samples with correction for multiple comparisons, while the differences between categorical variables were assessed via the χ^2^ test or the Fisher’s exact test. The observed significance level (*p*) was set at 5% (0.05) or less. In any case, this value is only meant to discriminate between large correlations in the sample and does not imply a proper inferential meaning. The answers provided by participating doctors regarding their attitudes towards drugs used for COVID-19, and their experiences with any suspect ADR, are expressed as % of responding doctors weighted for the number of COVID-19 patients included in the present study that each doctor cared for. Statistical calculations and graphs were performed using GraphPad Prism version 8.0.0 for Windows, GraphPad Software, San Diego, CA, USA, www.graphpad.com, accessed on 3 October 2022.

## 3. Results

### 3.1. Characteristics of COVID-19 Patients

From the 70 doctors volunteering during the selected period, 10 (14.3%) took part in the study (see Appendix B) and provided data concerning a total of 392 consecutive COVID-19 patients. The number of patients taken care of was: 13 (3.3% of total patients) in November 2020; 24 (6.1%) in December 2020; 51 (13%) in January 2021; 90 (23%) in February 2021; and 214 (54.6%) in March 2021. The characteristics of patients according to the disease stage are shown in Table 1.

The number of elapsed days, from the time of initial symptoms to the time patients were taken care of, was higher for those patients taken care of in stage 2a/2b (symptomatic with lung disease without/with desaturation) in comparison to those in stage 0/1 (asymptomatic/symptomatic without signs of lung disease). Patients presenting in stage 2b were older than those in stage 0/1/2a, and in turn, those in stage 1/2a were older than those in stage 0. The proportion of female patients was lower in stage 2b in comparison to all the other stages, however the difference was significant only with stage 1.

The proportion of overweight or obese patients increased from approximately two out of five in stage 0, to more than two out of three in stage 2b. The types of chronic comorbidities were mainly cardiovascular, metabolic and autoimmune, and their frequency increased from one out of five in stage 0, to more than seven out of ten in stage 2b. The number of chronic comorbidities was not different in patients presenting in the various stages. 

A positive result of the nasopharyngeal swab was available in 92.3% of cases, and in the remaining cases, the COVID-19 diagnosis was based on the results of later nasopharyngeal swabs and/or on clinical signs, symptoms and the disease course (including: in three cases, anosmia, and in one case, signs of lung disease leading to hospitalization).

### 3.2. Drug Prescriptions

Drugs prescribed for COVID-19 are shown in Table 2.

Vitamins and supplements, recommended by the IppocrateOrg Association guidelines since stage 0, were given to nearly all patients.

Among drugs recommended since stage 1, the most prescribed were aspirin and antibiotics. These drugs were increasingly prescribed throughout COVID-19 stages. Ivermectin and colchicine, also recommended since stage 1, were given to less than one in ten patients. Overall, one or more stage 1 drugs were increasingly prescribed throughout COVID-19 stages, from two out of five cases in stage 0 patients, to all the cases in stage 2b patients.

Enoxaparin, recommended since stage 2a, and glucocorticoids, recommended since stage 2b, were only occasionally used in those patients taken care of when in stage 0, but were given in 73–100% of patients taken care of when in stage 2b. Oxygen therapy, recommended in stage 2b only when SpO_2_ < 92% in ambient air, was never given in stage 0 patients, and given in one out of two stage 2b patients.

### 3.3. Outcomes

The outcomes of COVID-19 patients are shown in Table 3 according to the initial disease stage, and in Table 4 according to risk stratification [12].

The hospitalization of patients occurred in 25 (5.8%) cases. The hospitalization rate was lowest among patients taken care of when in stage 0 (one patient, 1.6%), and highest among those taken care of when already in stage 2b (six patients, 27.3%). The only hospitalized stage 0 patient was a 60-year-old female of normal weight and with cancer, who finally recovered with sequelae (joint pain). The stage 1 patients who underwent hospitalization, in comparison to those who were not hospitalized, did not differ in gender or weight but were older (on average, 60 years old vs. 47.1 years old, *p* < 0.05) and with more chronic comorbidities (1.3 vs. 0.4, *p* < 0.01). On the contrary, patients taken care of when in stages 2a and 2b who were hospitalized, did not differ, in terms of age, gender, weight or comorbidities, from patients taken care of in the same COVID-19 stage who did not undergo hospitalization (data not shown).

Altogether, 390 patients out of 392 (99.6% of total patients) recovered; in 88.7% of cases without sequelae, one patient (0.2%) was hospitalized and lost at follow up, and one patient (0.2%) died. This patient was a 72-year-old overweight male with cardiovascular disease, who had been taken into care on 6 March 2021, when in stage 2a of the disease, and who died after hospitalization.

The outcomes according to risk stratification (Table 4) show that hospitalization was more common in patients ≥ 60 (high risk A) or <60 but with comorbidities (high risk B), in comparison to patients < 60 and taken care of when in COVID-19 stage 0/1. The odds ratios for hospitalization (with 95% confidence intervals), in comparison to the low-risk group, are: 10.7 (2.6 to 48.7) for subjects in the high risk group A, and 10.1 (2.3 to 48.0) for subjects in the high risk group B. The use of a different age cut-off, e.g., 50 years (as suggested by other authors [13]), does not modify the results for the high risk group A and for the high risk group C, while in the high risk group B, hospitalizations fall from 11.8% to 5.6% and the difference with the low-risk group loses any statistical significance (not shown).

### 3.4. Doctors’ Attitudes towards the Use of Drugs in COVID-19

All the participating doctors responded in full to the questionnaire regarding attitudes concerning the choice, use and the possible ADR of drugs they had prescribed for COVID-19. The attitudes of doctors towards the use of drugs included in the IppocrateOrg Association guidelines, according to the various disease stages, are shown in Figure 1.

The main reasons for considering the use of aspirin in any COVID-19 stage include risk factors such as overweight or thrombophilia, occurrence of fever, inflammation and/or pain. The use of aspirin in stage 0 is considered, in particular, for patients who are not confident taking vitamins and supplements, or who cannot pay for these products.

Antibiotics are considered whenever fever and respiratory signs or symptoms occur, to prevent bacterial infections. Two doctors also mention the potential antiviral activity of azithromycin.

Regarding hydroxychloroquine, most doctors mention reasons not to prescribe this drug, including heart disease, stating they would always consider the drug in the absence of specific contraindications. In one case, this drug is considered as an alternative whenever ivermectin is not available.

Ivermectin is usually perceived as highly effective in COVID-19, and preferable in comparison to hydroxychloroquine because of its safe profile. Two doctors mention that this drug is often unavailable in Italian pharmacies, and one states that during the study period (1 November 2020–31 March 2021) s/he did not yet know about the usefulness of ivermectin in COVID-19 treatment.

Colchicine is usually considered as an alternative to ivermectin, or in association with ivermectin and hydroxychloroquine. Remarkably, two doctors state that they would choose colchicine whenever COVID-19 occurs with a severe headache, and one doctor would choose colchicine whenever signs of pericarditis occur.

The risk of thromboembolism is the main factor leading all doctors to consider enoxaparin for the treatment of COVID-19. Most doctors mention specific conditions such as overweight/obese or elderly subjects, hypertension, cardiac insufficiency and bedridden subjects. Three doctors also mention desaturation and respiratory symptoms/signs as reasons to consider the drug, and one states that s/he would also give enoxaparin in stage 1 whenever patients have positive anamnesis for previous thromboembolism.

Finally, glucocorticoids are generally considered whenever signs and symptoms, mainly fever, do not disappear after 4–5 days of treatment with other drugs and whenever SpO_2_ < 95%. The careful consideration of contraindications is mentioned by most doctors.

### 3.5. Perception and Reporting of Adverse Reactions to Drugs Used to Treat COVID-19

The perceived frequency of suspect ADR to drugs included in the IppocrateOrg Association guidelines is shown in Figure 2.

The drugs which are commonly estimated to cause ADR (frequency: 1–10%) are: antibiotics (43.2%), glucocorticoids (24.4%), hydroxychloroquine (18.7%), colchicine (17.7%), and aspirin (12.3%). Ivermectin and vitamins and supplements are associated with only uncommon ADR (frequency: <1%), and in just 25.4% and 17.7% of cases, respectively.

One doctor reported one grade 1 ADR (transient or mild discomfort (<48 h); no medical intervention or therapy required), namely urticaria (likely due to prescribed medicines). Three doctors reported, in total, eight grade 2 ADR (mild to moderate limitation in activity; some assistance may be needed; no or minimal medical intervention or therapy is required), namely: gastrointestinal discomfort with aspirin, antibiotics and glucocorticoids; hyperglycaemia with glucocorticoids; atrial fibrillation and diarrhoea with colchicine and hydroxychloroquine; allergy (likely due to prescribed medicines); gastrointestinal discomfort with aspirin, diarrhoea with hydroxychloroquine; and persistent hiccups with glucocorticoids.

## 4. Discussion

This is the first study to describe attitudes and behaviours of physicians taking care of COVID-19 outpatients, and the outcomes and safety of early treatment of COVID-19 outpatients in the real world. The outpatient treatment of COVID-19 is still a relatively poorly investigated area, and the few existing studies consist of trials of either individual therapeutics [14,15,16,17,18,19,20,21,22,23,24,25,26,27,28,29] or fixed associations [30]. The studies so far performed in Italy include: a retrospective assessment of a combination of indomethacin, low-dose aspirin, omeprazole, and a flavonoid-based food supplement, plus azithromycin, low-molecular-weight heparin, and betamethasone as needed, on the risk of hospitalization in 158 patients with early COVID-19 [31]; and a retrospective matched-cohort study comparing 90 patients with mild COVID-19 who were treated at home according to a recommendation algorithm, with 90 matched patients who were treated with the same drugs without considering the algorithm [32]. However, the pathophysiology of the SARS-CoV-2 infection is complex, and requires highly individualized therapeutic approaches, taking into account individual risk factors such as the patient’s age, sex and comorbidities, as well as the opportunity to combine direct antiviral therapy with anti-inflammatory and immunomodulating drugs, and antiplatelet/antithrombotic drugs [33]. For example, the I-MASK+ Early Outpatient Treatment Protocol for COVID-19, developed in 2020 by the Front Line COVID-19 Critical Care Alliance (FLCCC) and thereafter updated several times [34], includes: first-line agents (antivirals, such as ivermectin and hydroxychloroquine, antiseptics for nose and mouth wash, such as chlorhexidine, povidone-iodine and cetylpyridinium chloride, anticoagulants and immunomodulators, such as aspirin, vitamin D and melatonin, vitamins and supplements); second-line agents (such as antiandrogens, fluvoxamine and monoclonal antibodies); and third-line agents (such as corticosteroids). Nevertheless, the document makes clear that treatment for an individual patient should rely on the physician’s judgement, and that COVID-19 is a serious disease whose outcome depends on numerous factors including pre-existing conditions and the timing of treatment. Other outpatient treatment strategies have been proposed, which usually include vitamins and supplements, NSAID, corticosteroids and antibiotics in various combinations [35]. It is therefore of paramount importance to investigate the ways in which physicians manage COVID-19 patients in the real world, and the outcomes and safety of their choices. It is remarkable that physicians volunteering within the IppocrateOrg Association assistance network, and participating in our survey, chose drugs and drug combinations autonomously, and took only the best interests of the patient into account. Indeed, the guidelines developed and continuously updated by the IppocrateOrg Association represent a non-binding reference based on the best available scientific and clinical evidence, and are aimed at supporting the physicians’ choices in individual cases (S1); thus, the present study is the first genuine attempt to provide a detailed description of the early COVID-19 outpatient treatment in the real world.

The main result of the study concerns overall mortality: indeed, only one patient died from COVID-19 despite the management provided by the caring physician; therefore, the crude COVID-19 lethality in the present cohort of patients is 0.2%. According to official data, the COVID-19 lethality in Italy, which is standardized according to age and sex, was 6.6% during the first wave of infection from February to May 2020; 1.5% from June to September 2020; 2.4% in cases diagnosed in October 2020 [36]; and thereafter, 3% overall until 28 April 2021 [37]. According to the Worldometer website, the overall COVID-19 lethality in Italy ranged from 11.7 to 6.9% in November 2020; from 6.6% to 4.7% in December 2020; from 4.7% to 4.2% in January 2021; from 4.1% to 3.9% in February 2021; and from 3.8% to 3.6% in March 2021 (https://www.worldometers.info/coronavirus/country/italy/, accessed on 3 October 2022). Of course, no direct comparison is possible between our cohort and whole population data, due to the obvious differences in age and gender composition, and in established COVID-19 risk factors, including overweight/obesity and chronic comorbidities. Nevertheless, the mortality rate of 0.2% (one in 392 patients) observed in our cohort is clearly much lower than anyone would expect. In particular, the patient who died was taken care of in March 2021, together with a further 213 patients who eventually recovered, which leads to a crude COVID-19 lethality in patients taken care of in March 2021 of 0.5%, when the overall COVID-19 lethality in Italy was between 3% [37] and 3.8% (https://www.worldometers.info/coronavirus/country/italy/, accessed on 3 October 2022).

Remarkably, the care for the one deceased patient commenced when the patient was already in stage 2a of the disease, and his clinical profile presented several risk factors for a negative outcome, including: male gender, old age (77 years), overweight (BMI: 27.7), and chronic cardiovascular disease. He was treated with vitamins and dietary supplements, aspirin, antibiotics, hydroxychloroquine and glucocorticoids, and was thereafter admitted to hospital, where he eventually died due to COVID-19. No information is available regarding the patient’s hospital treatments. The factors of older age and male sex are indeed associated with a higher risk of death and hospitalization even in large cohorts of COVID-19 outpatients [38]_._ Indeed, overall, 113 patients were taken care of when already in COVID-19 stage 2a. In this subgroup, 19 (16.8%) were more than 65 years old and nine (8%) were more than 75; 28 (24.8%) were overweight and 20 (17.7%) were obese; 47 (41.6%) had at least one chronic comorbidity, and in particular, 35 (31%) had chronic cardiovascular diseases and 18 (15.1%) had two or more different chronic comorbidities. Nevertheless, only nine (8%) were admitted to hospital. Thus, despite the occurrence in this subgroup of several risk factors for severe COVID-19 and eventually a negative outcome, most patients recovered (111, 98.2%, since another patient was lost at follow up), suggesting an association between pharmacotherapeutic approaches chosen by physicians and favourable outcomes. Further support for this conclusion is provided by close consideration of the subgroup of patients taken care of when already in stage 2b of the disease: most of them (15, 68.2%) were more than 65 years old and 10 (45.5%) were more than 75; nine (40.9%) were overweight and three (13.7%) were obese; 16 (72.7%) had at least one chronic comorbidity, and nine (41%) had two or more different chronic comorbidities. Nonetheless, all of them successfully recovered. The indirect indication for the relevance of established risk factors for serious COVID-19 is provided by the analysis of our sample, based on risk stratification according to age, comorbidities and the COVID-19 stage at the time patients were taken care of [12]. The patients whose age was ≥60 or <60 but whose risk factors included comorbidities, had an average odds ratio of 10 for hospitalization in comparison to those without specific risk factors.

Our survey did not investigate the COVID-19 course in individual patients, however some data may be taken as an indirect indication of disease progression before recovery. The patient’s admission to hospital is the most obvious indicator of disease progression. In our survey, about one in four of the patients who were taken care of when already in stage 2b of the disease, were admitted to hospital before recovery; however, those patients who were taken care of in an earlier stage required hospitalization only seldomly (phase 2a: 8%) or very rarely (phase 1 and phase 0: 4.6% and 1.6%, respectively). The indirect indication of disease progression before recovery may also be obtained from information regarding drug prescriptions. Indeed, the need for oxygen therapy, which likely indicates progression up to blood desaturation (SpO_2_ < 92% in ambient air), never occurred in patients taken care of when in stage 0, very rarely occurred (2.5%) in patients in stage 1 (2.5%), and seldom occurred when patients were in stage 2a (9.7%), while 50% of stage 2b patients received oxygen therapy at home. Finally, according to the attitudes expressed by doctors participating in the survey towards the use of drugs in the various disease stages, it appears that antibiotics, enoxaparin and glucocorticoids would never be prescribed in stage 0 of the disease, and glucocorticoids would be never (50%) or only sometimes (50%) prescribed in stage 1. Thus, the use of each of these drugs in patients taken care of when in stage 0, likely indicates progression to symptomatic disease (stage 1 or beyond); in particular, the use of glucocorticoids in stage 0 or even stage 1 patients may indicate progression to stage 2. Indeed, patients taken care of when in stage 0 received antibiotics in 27.9% of cases, enoxaparin in 4.9% of cases and glucocorticoids in 11.5% of cases, while patients taken care of when in stage 1 received glucocorticoids in 25.5% of cases. Overall, such results suggest that a minor but nevertheless not negligible fraction of COVID-19 patients progress to symptomatic and eventually severe disease before recovery. Progression may occur more frequently in the presence of specific risk factors: indeed, our results show that the hospitalization rate is higher in patients taken care of when in stage 2b of the disease, and that patients taken care of when in stage 1 who were eventually hospitalized, were older and with more chronic comorbidities, in comparison to those who did not undergo hospitalization. Such observations further emphasize the need to treat COVID-19 as early as possible, and to carefully consider the presence of specific risk factors such as age and chronic comorbidities.

The doctors participating in our study were also asked to complete a short questionnaire investigating their propensity to use each specific drug in the various stages of COVID-19, and the relevant factors which eventually affected their decision. The questionnaire was also devised to collect information on any suspect ADR observed after the administration of anti-COVID-19 treatments. The doctors’ answers to the questionnaire are in line with pharmacotherapeutic choices in their patients (Table 2). Indeed, all the respondents agree regarding: the use of vitamins and supplements in any stage of the disease; the consideration of aspirin even in stage 0; the use of, in stage 1, not only stage 1 drugs but also enoxaparin and glucocorticoids (55% of respondents); and to use, in stage 2, antibiotics, enoxaparin and glucocorticoids. Such attitudes fit quite well with the choices recorded in the drug utilisation part of the study (Table 3): vitamins and supplements are prescribed to all patients; aspirin is prescribed to approximately one out of four patients taken care of when in stage 0; and enoxaparin and glucocorticoids are given to most patients in stage 2, but also to one in four or five patients in stage 1. The possibility that this finding also reflects disease progression before recovery has been discussed above.

Interestingly, the reasons provided for choosing or discarding specific drugs indicate a very good knowledge of the individual drugs’ indications and possible ADR. For instance, the main contraindications of hydroxychloroquine are carefully listed, conditions which recommend antibiotics or enoxaparin are well identified, as well as indications and contraindications of glucocorticoids. The doctors participating in the survey also document their remarkable experience with the possible ADR of drugs for COVID-19 (Figure 2). The perceived frequency of suspect ADR is mostly rated as “never” or “uncommon” (<1%), except for aspirin, hydroxychloroquine, colchicine and enoxaparin, where ADR are rated “uncommon” or “common” (1–10%) in 15.7–41.9% of cases and in 7.7–18.7% of cases, respectively, and antibiotics and glucocorticoids where ADR are rated “uncommon” or “common” in 17.7–50.9% of cases and in 23.7–43.2% of cases. Observed ADR were, however, few (there were only nine in the whole population) and mild (one) or mild to moderate (eight). Overall, this set of results suggest the good tolerability of drugs and drug combinations used for early COVID-19 treatment, as well as a very good knowledge of their characteristics by prescribing doctors.

Our study has many methodological limitations, of which we are well aware. The main limitation is its retrospective design. The physicians were asked to provide data regarding patients taken care of during an emergency situation, when they were volunteering with the overarching goal to care for patients in need and to save lives. For this reason, the collected data includes only essential information to obtain an overall picture of the COVID-19 patients’ clinical profiles and outcomes. Much more information is needed to describe in detail the clinical course of COVID-19 and its response to drug treatments: the time to recovery, drug doses, the time of administration, the nature, duration and outcomes of post-recovery sequelae, the reasons for hospitalization and the treatments received, etc. In addition, we lacked a control group; therefore, results could be compared only, and very cautiously, with official Italian whole population data referring to roughly the same time periods and concerning the COVID-19 lethality, but were not directly compared according to age range and sex, or for different outcomes, e.g., the admission to hospital. Finally, a small sample of doctors (10) took part in this first study; nevertheless, they were able to collect data concerning nearly 400 patients, which represents, so far, the largest sample of COVID-19 patients thoroughly investigated in Italy. All the participating doctors included only consecutive patients, thus minimizing any possible selection biases.

## 5. Conclusions

In conclusion, our study is the first to describe the attitudes and behaviours of physicians caring for COVID-19 outpatients, and the outcomes and safety of early treatment of COVID-19 outpatients in the real world. The COVID-19 lethality in our cohort of nearly 400 consecutive patients was 0.2% (only one patient died), thus the use of individual drugs and drug combinations as reported in our investigation was generally associated with favourable outcomes and safe results, as was also indicated by the few and mild reported ADR. The present study is part of an ongoing research program which aims to continuously collect and analyse all the information regarding patients cared for. Currently, information about COVID-19 patients cared for from April to July 2021 is being collected, while, commencing in August 2021, a specific procedure for the prospective collection of anonymized information has been included in the routinary activity of the IppocrateOrg Association’s organisational secretariat. This collected information will provide details regarding the outcomes and safety of the personalised early outpatient treatment for COVID-19 and will be developed and used by physicians volunteering within the IppocrateOrg Association. Meanwhile, we expect that the present evidence will be carefully considered by physicians caring for their COVID-19 patients as well as by political decision makers responsible for the management of the current global crisis.

## Figures and Tables

**Figure 1 jcm-11-06138-f001:**
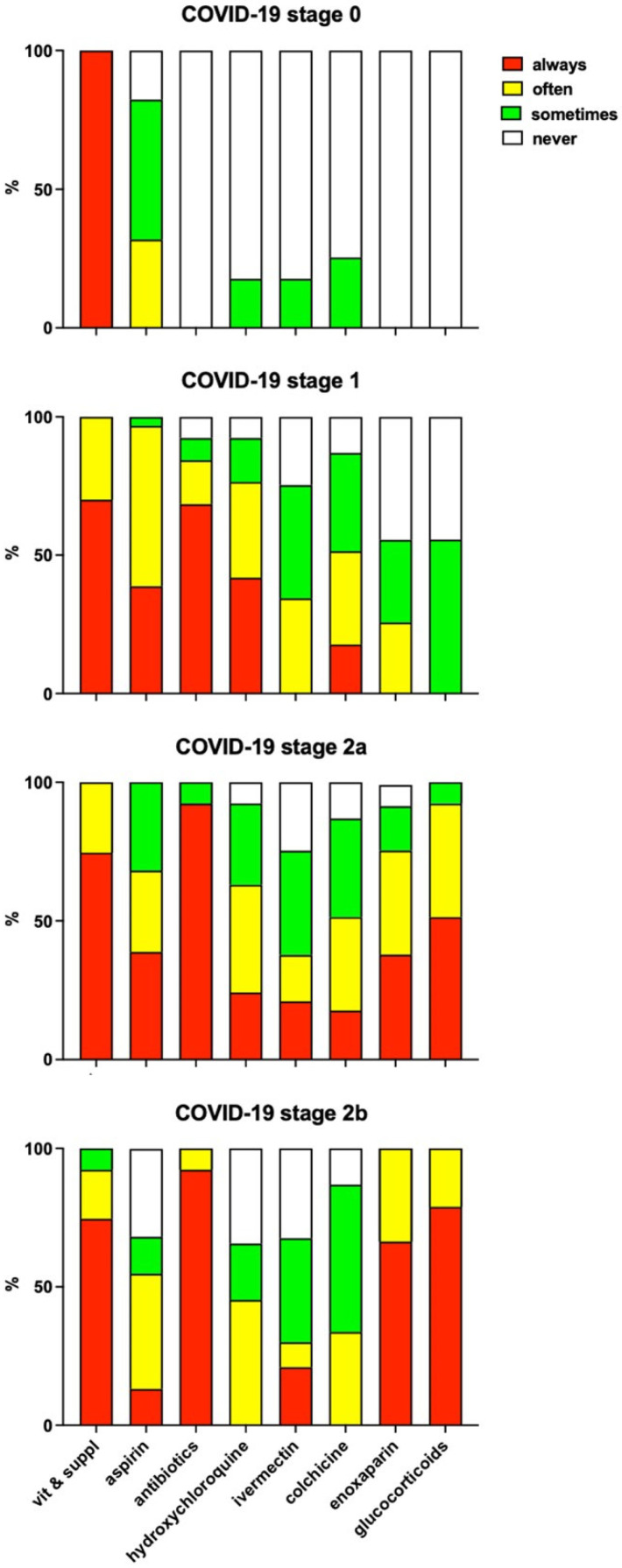
Attitudes of doctors towards the use of drugs, according to the various disease stages. Data is expressed as % of responding doctors weighted for the number of COVID-19 patients included in the present study that each doctor cared for.

**Figure 2 jcm-11-06138-f002:**
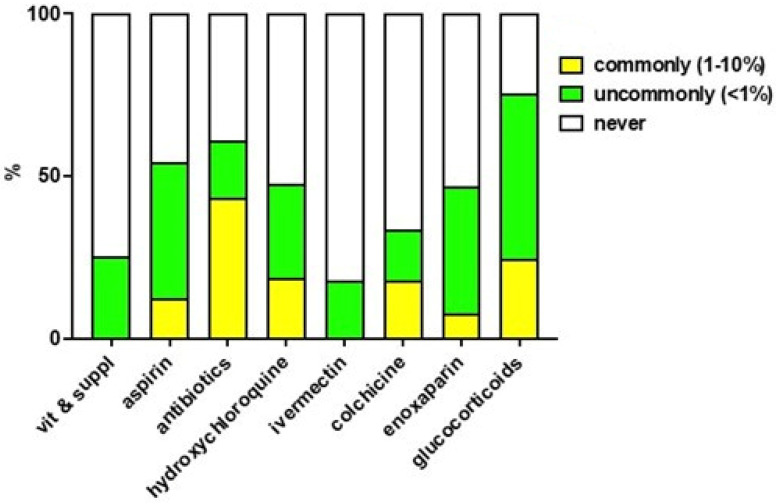
Perceived frequency of suspect ADR. Data is expressed as % of responding doctors weighted for the number of COVID-19 patients included in the present study for whom each doctor cared for.

**Table 1 jcm-11-06138-t001:** Characteristics of COVID-19 patients included in the study according to disease stage when care of patients commenced.

	COVID-19 Stage				Total
	0	1	2a	2b	
*n* (% of total)	61 (15.6)	196 (50.0)	113 (28.8)	22 (5.6)	392 (100)
days since symptoms began mean (min-max)	3.3 (0–7)	4 (0–20)	5.5 (1–20) *^,#^	5 (1–10) *	4.4 (0–20)
missing (*n*, % of total)	0 (0)	3 (1.5)	0 (0)	0 (0)	3 (0.8)
F/M (% F)	27/34 (44.3)	110/86 (56.1)	58/55 (51.3)	6/16 (27.3) ^#^	201/191 (51.3)
age (years) mean (min-max)	39.2 (0.5–89)	47.7 (1.4–97) *	51.3 (13–85) *	66.5 (44–90) *^,#,§^	48.5 (0.5–97)
age (*n*, % of total)					
<18	12 (19.7)	10 (5.1)	3 (2.7)	0 (0)	25 (6.4)
18–29	6 (9.8)	13 (6.6)	8 (7.1)	0 (0)	27 (6.9)
30–39	10 (16.4)	26 (13.3)	13 (11.5)	0 (0)	49 (12.5)
40–49	15 (24.6)	55 (28.1)	20 (17.7)	3 (13.6)	93 (23.7)
50–59	9 (14.8)	43 (21.9)	33 (29.2)	5 (22.7)	90 (23)
60–69	4 (6.6)	27 (13.8)	20 (17.7)	4 (18.2)	55 (14)
70–79	1 (1.6)	12 (6.1)	7 (6.2)	3 (13.6)	23 (5.9)
80–89	3 (4.9)	3 (1.5)	5 (4.4)	6 (27.3)	17 (4.3)
>89	0 (0)	1 (0.5)	0 (0)	1 (4.5)	2 (0.5)
missing	1 (1.6)	6 (3.1)	4 (3.5)	0 (0)	11 (2.8)
weight status *n* (% of total)					
Underw. (BMI <18.5)	3 (4.9)	7 (3.6)	4 (3.5)	0 (0)	14 (3.6)
Normal w. (BMI = 18.5–24.9)	21 (34.4)	77 (39.3)	37 (32.7)	5 (22.7)	140 (35.7)
Overw. (BMI = 25–29.9)	10 (16.4)	55 (28.1)	28 (24.8)	9 (40.9)	102 (26)
Obese (BMI > 30)	5 (8.2)	17 (8.6)	20 (17.7)	3 (13.7)	45 (11.5)
Underw. or Normal w./Overw. or Obese	24/15	84/72	41/48	5/12 *	154/147
missing	22 (36.1)	40 (20.4)	24 (21.3)	5 (22.7)	91 (23.2)
chronic comorbidities *n* (% of total)	12 (19.7)	62 (31.6)	47 (41.6) *^,#^	16 (72.7) *^,#,§^	137 (34.9)
cardiovascular	5 (8.2)	38 (19.4)	35 (31)	12 (54.5)	90 (23)
metabolic	5 (8.2)	25 (12.7)	14 (12.4)	8 (36.4)	52 (13.3)
respiratory	0 (0)	3 (1.5)	3 (2.6)	3 (13.7)	9 (2.3)
oncologic	2 (3.3)	1 (0.5)	4 (3.5)	3 (13.7)	10 (2.5)
neurologic	4 (6.6)	8 (4.1)	5 (4.4)	1 (4.5)	18 (4.6)
autoimmune	2 (3.3)	13 (6.6)	7 (6.2)	1 (4.5)	23 (5.9)
chronic comorbidities *n*/patient (mean ± SD)	1.3 ± 0.5	1.4 ± 0.7	1.4 ± 0.7	1.8 ± 0.8	0.5 ± 0.8
chronic comorbidities *n* (% of total)					
0	49 (80.3)	134 (68.4)	66 (58.4)	6 (27.3)	255 (66.1)
1	7 (11.5)	43 (21.9)	30 (26.5)	7 (31.8)	87 (22.2)
2	4 (6.6)	13 (6.6)	14 (12.4)	6 (27.3)	37 (9.4)
3	1 (1.6)	5 (2.5)	2 (1.8)	3 (13.7)	11 (2.8)
4	0 (0)	1 (0.5)	1 (0.9)	0 (0)	2 (0.5)
nasal swab *n* (% of total)					
positive	56 (91.8)	175 (89.2)	110 (97.4)	21 (95.5)	362 (92.3)
negative	0 (0)	7 (3.6)	0 (0)	0 (0)	7 (1.8)
not done	1 (1.6)	7 (3.6)	3 (2.6)	1 (4.5)	12 (3.1)
missing	4 (6.6)	7 (3.6)	0 (0)	0 (0)	11 (2.8)

* *p* < 0.05 vs. COVID-19 stage 0; ^#^ *p* < 0.05 vs. COVID-19 stage 1; ^§^ *p* < 0.05 vs. COVID-19 stage 2a. Statistical analysis is only meant to discriminate between large correlations in the sample and does not imply a proper inferential meaning. BMI = body mass index.

**Table 2 jcm-11-06138-t002:** Drugs prescribed throughout the treatment according to COVID-19 stage when patients were taken care of. Patients who were prescribed each drug or drug category are reported as absolute numbers and as percentage (in parentheses) of total patients in each COVID-19 stage.

	COVID-19 Stage				Total
	0	1	2a	2b	
*n* (% of total)	61 (15.6)	196 (50.0)	113 (28.8)	22 (5.6)	392 (100)
recommended since stage 0					
vitamins and supplements	61 (100)	196 (100)	108 (95.6)	22 (100)	387 (98.7)
recommended since stage 1					
Aspirin	16 (26.2)	139 (70.9)	89 (78.8)	15 (68.2)	259 (66.1)
Antibiotics	17 (27.9)	104 (53.1)	100 (88.5)	22 (100)	243 (62)
Hydroxychloroquine	4 (6.6)	65 (33.2)	38 (33.6)	9 (40.9)	116 (29.6)
Ivermectin	0 (0)	3 (1.5)	6 (5.3)	2 (9.1)	11 (2.8)
Colchicine	1 (1.6)	20 (10.2)	12 (10.6)	2 (9.1)	35 (8.9)
≥1 Stage 1 drug	26 (42.6)	174 (88.8)	111 (98.2)	22 (100)	323 (82.4)
recommended since stage 2a					
Enoxaparin	3 (4.9)	41 (20.9)	52 (46)	16 (72.7)	112 (28.6)
recommended since stage 2b					
Glucocorticoids	7 (11.5)	50 (25.5)	85 (75.2)	22 (100)	164 (41.8)
oxygen therapy	0 (0)	5 (2.5)	11 (9.7)	11 (50)	27 (6.9)

**Table 3 jcm-11-06138-t003:** Outcomes of COVID-19 patients included in the study according to disease stage when patients were taken care of.

	COVID-19 Stage				Total
	0	1	2a	2b	
*n* (% of total)	61 (15.6)	196 (50.0)	113 (28.8)	22 (5.6)	392 (100)
hospitalized *n* (% of total)	1 (1.6)	9 (4.6)	9 (8)	6 (27.3) *^,#,§^	25 (5.8)
recovered *n* (% of total)	61 (100)	196 (100)	111 (98.2)	22 (100)	390 (99.6)
without sequelae	50 (82)	173 (88.3)	100 (90.1)	19 (86.4)	342 (88.7)
with sequelae	11 (18)	23 (11.7)	11 (9.9)	3 (13.6)	48 (12.3)
deceased *n* (% of total)	0 (0)	0 (0)	1 (0.9)	0 (0)	1 (0.2)
missing *n* (% of total)	0 (0)	0 (0)	1 (0.9)	0 (0)	1 (0.2)

* *p* < 0.05 vs. COVID-19 stage 0; ^#^ *p* < 0.05 vs. COVID-19 stage 1; ^§^ *p* < 0.05 vs. COVID-19 stage 2a. Statistical analysis is only meant to discriminate between large correlations in the sample and does not imply a proper inferential meaning.

**Table 4 jcm-11-06138-t004:** Outcomes of COVID-19 patients included in the study according to risk stratification based on age, comorbidities and disease stage according to Derwand et al. [12]. The table does not include 11 patients with missing age (Table 1).

	COVID-19 Risk Group ^1^				Total
	Low	High A	High B	High C	
*n* (% of total)	154 (40.4)	97 (25.5)	68 (17.8)	62 (16.3)	381 (100)
hospitalized *n* (% of total)	2 (1.3)	12 (12.4) *	8 (11.8) *	3 (4.8)	25 (100)
recovered *n* (% of total)	154 (100)	96 (99)	67 (98.5)	62 (100)	379 (100)
without sequelae	138 (89.6)	84 (86.6)	49 (72.1)	60 (96.8)	331 (100)
with sequelae	16 (10.4)	12 (12.4)	18 (26.5)	2 (3.2)	48 (100)
deceased *n* (% of total)	0 (0)	1 (1)	0 (0)	0 (0)	1 (100)
missing *n* (% of total)	0 (0)	0 (0)	1 (1.5)	0 (0)	1 (100)

^1^ risk groups are defined as follows: high risk A: all patients with age ≥ 60; high risk B: patients with age < 60 and with one or more comorbidities; high risk C: patients with age < 60, no comorbidities, but initial COVID-19 stage 2a or 2b; low risk: patients with age < 60, no comorbidities, and initial COVID-19 stage 0 or 1. * *p* < 0.05 vs. COVID-19 stage 0. Statistical analysis is only meant to discriminate between large correlations in the sample and does not imply a proper inferential meaning.

## Data Availability

All data generated or analysed during this study is included in this published article and its Appendix A. Raw anonymized data is provided as Appendix A.

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
