# Peer review of "Early Outpatient Treatment of COVID-19: A Retrospective Analysis of 392 Cases in Italy"

_jcm, 2022, doi:10.3390/jcm11206138_

Round 1
Reviewer 1 Report
Referee Report
I congratulate the authors for successfully conducting and writing up this interesting study of early outpatient treatment protocols for COVID-19 used in Italy.
Two characteristics of this work that make it particularly interesting are that: (a) the authors report on the results of a sequenced multi-drug treatment protocol using objective endpoints (hospitalizations and deaths); (b) the authors have also studied and reported on the empirical data of how the multi-drug sequenced treatment protocol was actually implemented by the treating physicians, confirming that they are consistent with the IppocrateOrg guidelines; (c) although the prevention of hospitalizations appears to be sensitive with respect to the disease stage at which treatment is initiated, the prevention of mortality appears to be uniform across the board for all 4 stages of the disease considered by the authors. With regards to (a), COVID-19 is a complex multistage disease, and it is not reasonable to expect that a single drug will be effective against all stages of the disease. It follows that it is important to validate the effectiveness of a treatment protocol that uses multiple drugs in combination, and which also sequences the medications used in accordance with stage of the illness. The author's manuscript is one of very few studies that have explicitly attempted to validate a multi-drug sequenced outpatient treatment protocol against COVID-19. Furthermore, it is the only study to my knowledge that has looked at the empirical data of how such treatment protocols were implemented in the middle of an emergency by treating physicians. In this regard, they set an excellent example for future studies by other similar treatment groups from around the world.
There is some room for improving the manuscript in terms of data analysis of the treatment outcomes.
1. On table 1, I would recommend reporting the age demographic by using more granular intervals: <18, 19-29, 30-39, 40-49 50-59, 60-69, 70-79, 80-89, >90. The typical age thresholds used to define high-risk patients are typically set to age >50 or >60, and we see that the bulk of the patients are in the 46-65 interval that cuts across both thresholds. So, more granularity would be interesting. The proposed age bracket breakdown was previously used Million et al [1].
2. Table 3, showing the hospitalization and death endpoints against the stage level at which treatment was initiated, should be supplemented with additional tables that show the number of hospitalizations and deaths against the patient risk level.
There are several ways that the patient risk level can be quantified. One approach is to show the hospitalization and death outcomes in terms of the patient age brackets using the breakdown that I suggested above. Another approach is to use the risk stratification criteria by Zelenko [2] which break down the patients into three high-risk categories and one low-risk category as follows:
(1) High risk A: age >= 60
(2) High risk B: age < 60 and comorbidities >= 1
(3) High risk C: age <= 60 and comorbidities = 0 and (stage 2a OR stage 2b)
(4) Low risk: None of the above
It is also interesting to look at the sensitivity of the results if one uses age 50, recommended by Ref.[3] instead of age 60 as the age threshold in the Zelenko definition of high risk patients.
Reporting on the hospitalizations and deaths for treated patients against age brackets (the approach used by Ref. [1]) and also against the high-risk and low-risk patient definition used by Zelenko (with an age 50 and/or age 60 threshold), may produce interesting results, and I encourage the authors to report these results in an updated version of the current manuscript. In particular, it will be interesting to identify the patient group characteristics that make them susceptible to hospitalization, in spite of early outpatient treatment, and quantify the hospitalization risk specific to these groups.
I commend the efforts and the courage of the Italian doctors that are part of the IppocrateOrg group. It is important that the real world data that has been accumulated, as a result of their efforts, be published and analyzed to the extent possible. The recommended revisions are aimed towards that goal.
References
[1] M. Million, J-C. Lagier, H. Tissot-DuPont, I. Ravaux, C. Dhiver, C. Tomei, N Cassir, L. DeLorme, S. Cortaredona, S. Gentile, E. Jouve, A. Giraud-Gatineau, H. Chaudet, L. Camoin-Jau, P. Colson, P. Gautret, P-E. Fournier, B. Maille, J-C. Deharo, P. Habert, J-Y. Gaubert, A. Jacquier, S. Honore, K. Guillon-Lorvellec, Y. Obadia, P. Parola, P. Brouqui, D. Raoult, IHU COVID-19 Task Force. "Early Treatment with Hydroxychloroquine and Azithromycin in 10,429 COVID-19 Outpatients: A Monocentric Retrospective Cohort Study", Reviews in Cardiovascular Medicine 22 (2021), 1063-1072
[2] M. Scholz, R. Derwand, V. Zelenko. "COVID-19 outpatients - early risk-stratified treatment with zinc plus low dose hydroxychloroquine and azithromycin: a retrospective case series study", International Journal of Antimicrobial Agents 56 (2020), 106214
[3] P.A. McCullough, P.E. Alexander, R. Armstrong, C. Arvinte, A.F. Bain, R.P. Bartlett, R.L. Berkowitz, A.C. Berry, T.J. Borody, J.H. Brewer, A.M. Brufsky, T. Clarke, R. Derwand, A. Eck, J. Eck, R.A. Eisner, G.C. Fareed, A. Farella, S.N.S. Fonseca, C.E. Geyer, Jr., R.S. Gonnering, K.E. Graves, K.B.V. Gross, S. Hazan, K.S. Held, H. Thomas Hight, S. Immanuel, M.M. Jacobs, J.A. Ladapo, L.H. Lee, J. Littell, I. Lozano, H.S. Mangat, B. Marble, J.E. McKinnon, L.D. Merritt, J.M. Orient, R. Oskoui, D.C. Pompan, B.C. Procter, C. Prodromos, J.C. Rajter, J-J. Rajter, C. V.S. Ram, S.S. Rios, H.A. Risch, M.J.A. Robb, M. Rutherford, M. Scholz, M.M. Singleton, J.A. Tumlin, B.M. Tyson, R.G. Urso, K. Victory, E.L. Vliet, C.M. Wax, A.G. Wolkoff, V. Wooll, V. Zelenko. "Multifaceted highly targeted sequential multidrug treatment of early ambulatory high-risk SARS-CoV-2 infection (COVID-19)", Reviews in Cardiovascular Medicine 21 (4) (2020), 517-530
Reviewer 2 Report
Why is this information being presented in Sept. 2022? What is the value this late in the course of the pandemic? Authors should clarify the contributing physician sample. Were they involved in developing guidelines? In Study procedures, it states that physicians were ‘invited to provide’. How much of the requested data must be complete for a person to be included in the analysis? How was the missing information managed from a deign standpoint? From a statistical standpoint? The information provided in the statistical analysis is lacking. How many comparisons were done? What were the two normally distributed independent samples used for each t-test? Should the p-value be adjusted to account for the number of comparisons? The results section implies more analyses were done than the ones mentioned in the statistical analysis section. The large number of comparisons performed are not supported by the sample size. The number of comparisons performed between individual groups (for example the hospitalizations) are reported without the larger group level comparison p-value reported and the number in the individual stages are too small to support additional comparisons. What about data from patients who were not treated or denied the recommended treatment? I assume the vast majority of those people survived, were not hospitalized, and had no long-term issues.
Major
This sample is subject to bias due to the very low participation rate of the physicians (14.3%). All of the data came from only 10 physicians. What percentage of patient data was available compared to the total patient volume treated by the 70 physicians? The results of the comparisons for patient characteristics should be reported as a final column in Table 1. Cherry-picking results in the text is not helpful and adding it to the table would eliminate some text in the results section. The level of reporting (individual group differences like the results for age across groups) was not supported by the description on the statistical analysis section. Same issue for Tables 2 and 3. If comparisons were performed, they should be included as the last column in both tables. Figure 1 has no N. Did all 10 physicians answer all questions? Most of the text in the results section for the survey is not needed. The opinion of 10 physicians before March 2021 with no outcomes data is not helpful, especially in 2022. Is Fig. 2 perceived or observed ADR? There is no analysis looking at outcome by treatment. Focusing one the fact that only 1 person from the data collected died adds nothing to the literature. The authors are missing data from 60 physicians on an unknown number of patients just based on the POSSIBLE data that could have been collected for these analyses based on the defined population. RECOVERY RATE WITHIN A GROUP DOES NOT SUGGEST EFFECTIVENESS. There isn’t even a discussion of this considering the recovery rate with no treatment. With such a high recovery rate, you could look at almost any variable (via correlation, retrospectively) that is common among the population and find that it correlates with recovery. Authors falsely interpret progression data as being related to treatment even while acknowledging the risk factors among those that progressed. None of the evidence presented suggests that people who were treated did any better than those who were not treated.
Minor Comments
Line 93 – would delete ‘cured’ by doctors through teleassistance. Methods are not proven and cure rate without treatment is above 95%.
Again – Table 2 should not use the word ‘cure’.
Reference missed:
Application of an evidence-based, out-patient treatment strategy for COVID-19: Multidisciplinary medical practice principles to prevent severe disease.
Frohman EM, Villemarette-Pittman NR, Rodriguez A, Glanzman R, Rugheimer S, Komogortsev O, Zamvil SS, Cruz RA, Varkey TC, Frohman AN, Frohman AR, Parsons MS, Konkle EH, Frohman TC.J Neurol Sci. 2021 Jul 15;426:117463. doi: 10.1016/j.jns.2021.117463. Epub 2021 Apr 20.PMID: 33971376
Round 2
Reviewer 2 Report
The authors have attempted to address concerns raised by reviewers. The value to the scientific literature is limited, but I will keep my comments targeted toward concerns regarding conclusions or assumptions made by the authors.
The updated tables are helpful to characterizing the sample.
Table 1 notes that just shy of 1/4 of the 2b sample is missing BMI, yet the authors highlight differences in weight from stage 0 to stage 2b. If all 5 missing data points in 2b are normal weight, the difference is negligible. That also assumes that the missing 1/3 of the stage 0 sample is distributed similarly to the rest of the sample, which is unknown.
Table 4 shows outcomes by risk and authors report increased odds of hospitalization based on higher risk. They state in their response that they do not connect outcome to treatment or progression to treatment, but here are examples:
"Thus, despite the occurrence in this subgroup of several risk factors for severe COVID-19 and eventually a negative outcome, most patients recovered (111, 98.2%, since another patient was lost at follow up), suggesting the favourable outcomes of the pharmacotherapeutic approaches chosen by physicians" lines 365-367
"Further support to this conclusion is provided by close consideration of the subgroup of patients taken care of when already in stage 2b....." line 368
And in the section regarding meds prescribed "Such observations further emphasize the need to treat COVID-19 as early as possible, and to carefully consider the presence of specific risk factors such as age and chronic comorbidities." lines 405-407
Authors say they do not emphasize mortality without noting the % recovered for those without treatment (essentially the same), but page 11, line 333 states "The main result of the study is about overall mortality: indeed, only one patient died of COVID-19 despite the management provided by the caring physician" and spend a page discussing mortality and the reported 'lower' rate in their study. For example, "Nevertheless, the mortality rate of 0.2% (one in 392 patients) observed in our cohort is clearly much lower than anyone would expect." lines 345-346
In Conclusions, they again mention lethality and effectiveness "COVID-19 lethality in our cohort of nearly 400 consecutive patients was 0.2% (only one patient died), thus the use of individual drugs and drug combinations as reported in our investigation resulted effective and safe, as also indicated by few and mild reported ADR." lines 458-460
And efficacy again in Conclusions "Collected information will provide details about efficacy and safety" line 466
ADR was perceived, which the authors clarify in the figure, but in the text state repeatedly "observed ADR" or "observed frequency". These are estimates from memory from 10 doctors.
